# Semi-Automatic Measurement of Fetal Cardiac Axis in Fetuses with Congenital Heart Disease (CHD) with Fetal Intelligent Navigation Echocardiography (FINE)

**DOI:** 10.3390/jcm12196371

**Published:** 2023-10-05

**Authors:** Alexander Weichert, Michael Gembicki, Jan Weichert, Sven Christian Weber, Josefine Koenigbauer

**Affiliations:** 1Center for Prenatal Diagnosis and Women’s Health, 10961 Berlin, Germany; weichert@bergmannstrasse102.de; 2Departments of Obstetrics and Gynecology, University of Schleswig-Holstein, Campus Lübeck, 23538 Lübeck, Germany; michael.gembicki@uksh.de (M.G.); jan.weichert@uksh.de (J.W.); 3Department of Pediatric Cardiology, Charité—Universitätsmedizin Berlin, 13353 Berlin, Germany; sven-christian.weber@dhzc-charite.de; 4Department of Obstetrics, Charité—Universitätsmedizin Berlin, 10117 Berlin, Germany

**Keywords:** fetal cardiac axis, FINE, fetal intelligent navigation echocardiography, CHD, congenital heart disease, AI

## Abstract

Congenital heart disease (CHD) is one of the most common organ-specific birth defects and a major cause of infant morbidity and mortality. Despite ultrasound screening guidelines, the detection rate of CHD is limited. Fetal intelligent navigation echocardiography (FINE) has been introduced to extract reference planes and cardiac axis from cardiac spatiotemporal image correlation (STIC) volume datasets. This study analyses the cardiac axis in fetuses affected by CHD/thoracic masses (*n* = 545) compared to healthy fetuses (*n* = 1543) generated by FINE. After marking seven anatomical structures, the FINE software generated semi-automatically nine echocardiography standard planes and calculated the cardiac axis. Our study reveals that depending on the type of CHD, the cardiac axis varies. In approximately 86% (471 of 542 volumes) of our pathological cases, an abnormal cardiac axis (normal median = 40–45°) was detectable. Significant differences between the fetal axis of the normal heart versus CHD were detected in HLHS, pulmonary atresia, TOF (*p*-value < 0.0001), RAA, situs ambiguus (*p*-value = 0.0001–0.001) and absent pulmonary valve syndrome, DORV, thoracic masses (*p*-value = 0.001–0.01). This analysis confirms that in fetuses with CHD, the cardiac axis can significantly deviate from the normal range. FINE appears to be a valuable tool to identify cardiac defects.

## 1. Introduction

Congenital heart disease is the most common organ-specific fetal malformation with an incidence of 8–10 per 1000 (0.8–1%) liveborn [1,2]. Children with CHD are associated with increased short-term and long-term morbidity and mortality [3,4,5,6]. About 50–60% of patients with CHD will require cardiac surgery [7]. Early diagnosis of CHD during the prenatal period provides information for the parents to enable counseling and decision-making on the fetal outcome and enables diagnostic tests (e.g., genetic, infectious, metabolic) to rule out an underlying cause for the specific CHD. Interdisciplinary counseling, including neonatologists, pediatric cardiac surgeons, and pediatric cardiologists, as well as psychologists and social workers, can be initiated upon early detection of CHD. Furthermore, the delivery of the fetus affected by CHD can be planned in a perinatal center with a pediatric cardiac surgery unit, which, therefore, improves neonatal survival [8].

The development of echocardiography was initiated in the 1960s. In 1954, Edler and Hertz from the University of Lund, Norway, demonstrated cardiac time-motion or M-mode recordings via ultrasound [9]. In 1972, Nanda and Gramiak from the University of Rochester were able to depict the pulmonary valve through M-mode, which led to an exponential development of echocardiography and the detection of CHD [10]. In the 1970s, fetal application in echocardiography advanced significantly. Kleinman established M-mode and B-mode examinations of the fetal heart, effectively diagnosing cases with CHD [11]. Over the next decades, fetal echocardiography experienced further advances by applying color Doppler, three- and four-dimensional imaging techniques, as well as computerized imaging [12]. Today, fetal echocardiography is an essential component of the fetal anomaly scan and finds a routine application during the first, second, and third-trimester scans.

The visualization of the four-chamber view (4CV) in fetal echocardiography is crucial for several reasons [13]. Firstly, it allows a comprehensive assessment of the fetal heart’s structure and functionality. It provides a clear visualization of the atria, ventricles, AV valves, and the inter-ventricular septum, thus enabling the detection of up to 42.8% of cardiac anomalies [14]. These may include congenital heart defects, such as ventricular asymmetry and septal defects, which can significantly impact the fetal heart’s functionality [15,16]. Secondly, the four-chamber view (with additional Doppler interrogation) might aid in evaluating the cardiac rhythm and the flow of blood within the heart chambers [13,17]. By observing the movements of the atrioventricular valves and the ventricular contractions, it becomes possible to assess the synchronization and efficiency of the cardiac cycle. Additionally, the visualization of the blood flow through the chambers aids in identifying any obstructions or abnormalities affecting the circulation, such as stenosis or regurgitation of valves. Lastly, the four-chamber view provides an opportunity to assess the overall development of the fetal heart. It allows for the measurement of key parameters like the size and contractility of all chambers, the thickness of the ventricular walls, and the integrity of normal anatomical structures like the interatrial and interventricular septa. These measurements will give important information regarding appropriate cardiac growth and development. Furthermore, the 4CV is instrumental in determining the position and orientation of the fetal heart in relation to the lungs within the chest cavity [18,19]. This information is crucial for diagnosing conditions like dextrocardia, where the heart is located on the right side of the chest instead of the left. Understanding the heart’s position as well as the fetal cardiac axis is essential to facilitate accurate diagnosis and planning further management or interventions. An abnormal cardiac axis can be associated with CHD, and the likelihood of an abnormal cardiac axis depends on the CHD type [20,21].

The detection rate of congenital heart defects (CHDs) reportedly depends on several factors, and despite advances that have been made in recent years, it remains challenging to achieve satisfying detection rates. Possible explanations include a lack of routine prenatal screening, either by access to prenatal care or the absence of standardized protocols for routine prenatal screening. Additionally, ultrasound is an operator-dependent technology. The accuracy of CHD detection can depend on the expertise and experience of the healthcare professional performing and interpreting the diagnostic tests [22]. A recent meta-analysis revealed a detection rate of CHD of 45.1% in an unselected population, with an even higher rate among univentricular defects and heterotaxy of above 85% [23]. Rural residence and, therefore, limited access are associated with a decreased rate of prenatal diagnosis of major CHD and an increased risk of late diagnosis (≥22 gestational weeks) [24].

Still, some CHD cases remain undetected. The most commonly undiscovered CHD are conotruncal defects with a normal four-chamber view (TGA, Fallot’s tetralogy, DORV, truncus arteriosus) [25]. With the advent of high-resolution ultrasound systems and AI-driven software solutions in the last decade, it has been suggested that four-dimensional ultrasound with spatiotemporal image correlation (STIC) might increase the detection rate of CHD [26,27,28]. Multiple computerized aids were developed to improve fetal echocardiography. Volume measurements applying 4D ultrasound facilitate virtual organ computer-aided analysis (VOCAL). With VOCAL ventricular volume, as well as total systolic volume, ejection fraction and cardiac output become assessable [29,30]. With SonoAVC or M-STIC, the AV valve plane can be analyzed, assessing the mitral and/or tricuspid annular plane systolic excursion (MAPSE/TAPSE) [31,32]. ESTIC (electronic spatiotemporal image correlation) helps to optimize the acquisition time and image quality in 4D echocardiography [33]. Tomographic ultrasound imaging (TUI) or multi-slice view (MSV) creates image sections of the same plane, but it is limited to the actual plane and fails in cases where the cardiac axis is oriented differently [34,35].

Fetal intelligent navigation echocardiography (FINE) is an even more advanced approach: the software generates a virtual map of the STIC volume dataset, creating nine standard fetal echocardiographic views [36,37,38,39]. Additionally, automated measurements and calculations, such as the cardiac axis, can be performed. The abnormal cardiac axis is often the first hint of an underlying cardiac anatomic pathology. In complex CHD, including dextrocardia and situs solitus or left heart abnormalities, FINE can detect multiple abnormalities and define the complex anatomic relationship [28,40]. FINE has a high detection rate with a sensitivity of 98% and a specificity of 93% [40,41]. In the analysis from Yeo et al., CHD detected by FINE completely matched 74% of cases, with minor discrepancies in 12% and major discrepancies in 14% [42]. FINE has been demonstrated to be useful in the assessment of CHD [43]. Recently, FINE has been improved with a very rapid static volume acquisition, including a high frame rate or precise control of the cardiac plane along the x, y and z axis [42].

In this multicenter analysis, we aimed to scrutinize the utility of FINE in regard to the fetal cardiac axis in fetuses with CHD and healthy fetuses.

## 2. Materials and Methods

In this study, STIC volumes of fetal echocardiography of cases with CHD, thoracic masses, or situs ambiguus were identified and analyzed retrospectively between 2016 and 2018. As a control group, volumes of normal fetal hearts were utilized. STIC works by acquiring a series of 2D ultrasound images over several cardiac cycles. These images are obtained by volume ultrasound probe to capture different views of the fetal heart. The timing of the image acquisition is synchronized with the fetal heart rate to ensure consistency. Once the series of images is acquired, sophisticated software algorithms are applied to analyze and correlate the images. The software tracks and matches corresponding structures and features in each frame, taking into account their spatial position and temporal changes. By analyzing these correlations, a dynamic 4D model of the fetal heart and its movements is reconstructed. The FINE technique begins with a standard STIC volume acquisition. The operator identifies seven standardized landmarks (therefore, semi-automatic measurement). Based on these findings, a virtual map of the volume dataset is created. Finally, FINE semi-automatically generates and displays the nine standard fetal echocardiographic views by applying intelligent navigation technology to STIC datasets [36,37,38,39]. The FINE software can also perform automated measurements and calculations, such as the cardiac axis. The volumes were obtained by physicians applying a WS80A Elite US system (Samsung Medison, Seoul, Republic of Korea). The datasets were acquired from an apical four-chamber view using a mechanical convex transducer (1–8 MHz) by applying transverse sweeps through the fetal chest. The acquisition time was 10 to 12 s, with an acquisition angle from 20° to 35°.

The patient’s consent was obtained in written form. Ethical approval was received from the ethics committee of Charité University Hospital on 21 April 2016 (EA2/066/16). CHD cases that were included in this study were hypoplastic left heart syndrome (HLHS, *n* = 157), atrioventricular septal defect (AVSD, *n* = 75), double outlet right ventricle (DORV, *n* = 87), as well as other major CHD and rare CHD. Additionally, cases with rhabdomyoma (*n* =9), thoracic masses (*n* = 11), and situs ambiguus (*n* = 15) were included. Initially the STIC volume was generated. Following the generation of seven anatomical planes, the FINE software semi-automatically created nine echocardiography standard planes based on the ISUOG and AIUM guidelines. Additionally, the cardiac axis was calculated automatically. The STIC images were generated at the two prenatal centers (Lübeck and Berlin, Germany). Statistical analysis was conducted using GraphPad Prism 9 for Mac (Version 9.51, GraphPad Software Inc., La Jolla, CA, USA), GraphPad QuickCalcs (GraphPad Software Inc., La Jolla, CA, USA), and Microsoft Excel for Mac (Version 16.71, Microsoft Corp., Redmond, WA, USA). Descriptive statistics, *t*-tests, McNemar tests, and Kruskal–Wallis tests were applied. A *p*-value of < 0.05 was assumed to be significant.

## 3. Results

Five hundred forty-five volumes of CHD or thoracic masses were included and compared to 1543 normal fetal hearts. The median of the normal fetal cardiac axis is between 40 and 45° [18,44] (Figure 1). The different types of CHD and the thoracic masses/anomalies are listed in Table 1, as well as the distribution among the measured volumes (mean, median, range). Among the 545 volumes obtained from CHD or thoracic masses cases, there were 7 volumes of the absent pulmonary valve, 5 of aortic stenosis, 5 of atrioseptal defect (ASD), 75 of atrioventricular septal defect (AVSD), 20 of coarctation aortae, 20 of complex CHD, 2 of double aortic arch (DAA), 87 of double outlet right ventricle (DORV), 157 of hypoplastic left heart syndrome (HLHS), 27 of hypoplastic right heart syndrome (HRHS), 1 of interrupted aortic arch (IAA), 19 of other CHD, 19 of pulmonary atresia, 12 of right aortic arch (RAA), 5 of rhabdomyoma, 15 of situs ambiguus (heterotaxy), 30 of transverse aortic constriction (TAC), 21 of transposition of the great arteries (TGA), 25 of tetralogy of Fallot (TOF), as well as 11 of thoracic masses. In 86% (471 out of 545 volumes), an abnormal cardiac axis was detected. Median and interquartile ranges of the cardiac axis were assessed by the FINE software (Figure 1, Figure 2, Figure 3 and Figure 4). Of note, situs ambiguous (heterotaxy) refers to the abnormal position of thoracic and visceral organs but does not necessarily imply a specific type of CHD. The cardiac axis in CHD cases was compared to the control. In CHDs such as absent pulmonary valve, aortic stenosis, ASD, pulmonary atresia, TOF, and complex CHD, the cardiac axis was >45°. In CHD such as RAA, situs ambiguus the heart axis appeared to be <20°. CHDs such as HRHS, situs ambiguous, and TGA displayed a high range of the cardiac axis. Kruskal–Wallis test showed a significant deviation from the normal fetal heart axis in cases of HLHS, pulmonary atresia, TOF (*p*-value < 0.0001), RAA, situs ambiguus (*p*-value = 0.0001–0.001) absent pulmonary valve syndrome, DORV, thoracic masses (*p*-value = 0.001–0.01).

Figure 2 visualizes the structured analysis of all nine planes in a fetus with normal cardiac anatomy: three vessels and trachea view, four-chamber view, five-chamber view, left ventricular outflow tract, the short axis of the great vessels (right ventricular outflow tract), abdomen view, ductal arch, aortic arch and venae cavae. The cardiac axis is calculated at 46.6°. In contrast, Figure 3 depicts a case of double inlet left ventricle (DILV) with pulmonary valve atresia with a cardiac axis of 67.5°. In this case, FINE is able to visualize the pathology that affects different cardiac planes in one examination. The three vessels and trachea view show the absence of the truncus pulmonalis. The four-chamber view reveals the non-appearance of the ventricular septum. Additionally, the right ventricular outflow tract and ductal arch indicates further the pulmonary valve atresia and stenosis of the truncus pulmonalis. Simply looking at the four-chamber views and cardiac axis obtained by FINE (Figure 4) underlines how effective and advanced this approach is. In Figure 4, various CHD and thoracic anomalies are depicted in comparison to the normal fetal heart—in most cases, the abnormal cardiac axis becomes apparent.

## 4. Discussion

The present study is able to demonstrate that FINE is a good tool to assess the fetal cardiac axis during a standardized semi-automated fetal echocardiography (Figure 2, Figure 3 and Figure 4). Fetal intelligent navigation echocardiography (FINE) has been developed to interrogate STIC volume datasets by applying an “intelligent navigation” technology. It facilitates the automatic display of nine standard fetal echocardiography views that are required to diagnose most cardiac defects (Figure 2, Figure 3 and Figure 4). This analysis highlights the importance of assessing the fetal cardiac axis and depicts the high range and or/deviation of the cardiac axis in fetuses with CHD or thoracic anomalies compared to healthy control. So far, this is the first study which focuses on the cardiac axis in a wide range of CHD and other thoracic pathologies. The fetal cardiac axis in normal controls ranges mainly between 40 and 45° (normal range 15–55°), which was confirmed in our healthy controls (Figure 1). In 86% (471 of 542 volumes) of our pathological cases, an abnormal cardiac axis was detected. Interestingly, CHDs such as absent pulmonary valve, aortic stenosis, ASD, pulmonary atresia, TOF, and complex CHD display a cardiac axis of >45°. In CHD such as RAA, situs ambiguus the heart axis appears to be <20°. It is of note that CHD, like HRHS, situs ambiguous, and TGA display a high range of the cardiac axis. Significant differences between the fetal axis in the normal heart versus CHD were detected in HLHS, pulmonary atresia, TOF (*p*-value < 0.0001), RAA, situs ambiguus (*p*-value = 0.0001–0.001), and absent pulmonary valve syndrome, DORV, thoracic masses (*p*-value = 0.001–0.01).

Our data are in accordance with other recent studies, which were able to demonstrate that FINE is a good tool to accurately diagnose CHD. An abnormal fetal cardiac axis can point towards CHD and promote an early diagnosis [40,41,42,43]. A recent analysis was able to demonstrate that the cardiac axis is significantly different from the normal axis in conotruncal anomalies (DORV, TAC, and TOF) [45]. Additionally, the study showed that in fetuses with TGA, the fetal cardiac axis does not differ compared to the normal axis. The group suggests the evaluation of the fetal cardiac axis, especially in screening for conotruncal anomalies. Several studies were able to demonstrate that the evaluation of the fetal cardiac axis can be an additional helpful tool in the prenatal diagnosis of CHD. Zhao et al. were able to reveal a correlation between neonatal death and an abnormal fetal cardiac axis in fetuses with tetralogy of Fallot [46]. They argued that an abnormal cardiac axis is associated with pulmonary atresia, right-sided aortic arch, and, therefore, a more complicated form of tetralogy of Fallot, which explains the higher risk of adverse neonatal outcome. A recent analysis suggests that a fetal cardiac axis is an additional tool for screening for CHD and fetal aneuploidy during the first-trimester scan [47]. They calculated a sensitivity of the fetal cardiac axis of 50.0% for CHD and 41.2% for fetal aneuploidy. Additionally, the cardiac axis can also be measured with fetal MRI when dedicated ultrasound and echocardiography are technically limited due to different aspects (e.g., BMI, multiples, fetal position). Liu et al. demonstrated that the cardiac axis measurement by fetal cardiac MRI is coherent to the sonographic cardiac axis evaluation [21].

An abnormal fetal cardiac axis should raise concerns for CHD or other fetal anomalies. FINE might be a reliable method to facilitate prenatal diagnosis of CHD and benefit counseling parents and professionals on decision-making on the pregnancy and follow-up of the fetus. With an early and accurate diagnosis, FINE might improve neonatal outcomes. It is possible to simply analyze the fetal cardiac axis with FINE. This tool is applicable to remote areas where sonographic training is limited [22].

This study is limited by the retrospective nature of the analysis and a selection bias, as not all patients who presented at the centers during the study period obtained a 3D ultrasound, including STIC volume. The volumes were generated by ultrasound experts, so the image and data quality are high.

## 5. Conclusions

This study reveals that the cardiac axis very likely deviates from CHD and varies depending on the type of underlying cardiac pathology. Especially in fetuses with HLHS, pulmonary atresia, TOF (*p*-value < 0.0001), RAA, situs ambiguus (*p*-value = 0.0001–0.001), and absent pulmonary valve syndrome, DORV, thoracic masses (*p*-value = 0.001–0.01) the fetal cardiac axis was significantly different from normal heart. An abnormal fetal cardiac axis should raise the suspicion of an underlying CHD.

The results confirm that FINE is a valuable tool for accurate, standardized detection and identification of CHD. Beyond that, our data show that combining the results with a semi-automatic assessment of the cardiac axis might improve the detection rate of fetuses with CHD. The evaluation of the fetal cardiac axis is an essential part of fetal echocardiography as it can facilitate the detection of CHD.

The information yield from FINE depends on the examiner’s scanning skills and image optimization, as well as the examiner’s expertise and experience in this field. Therefore, further teaching of FINE in prenatal diagnosis worldwide to improve the skills and experience of the examiners improves early detection and assessment of CHD and could possibly improve fetal outcomes. In conclusion, FINE will aid in improving the prenatal diagnosis and assessment of CHD as well as other stressful conditions in utero.

## Figures and Tables

**Figure 1 jcm-12-06371-f001:**
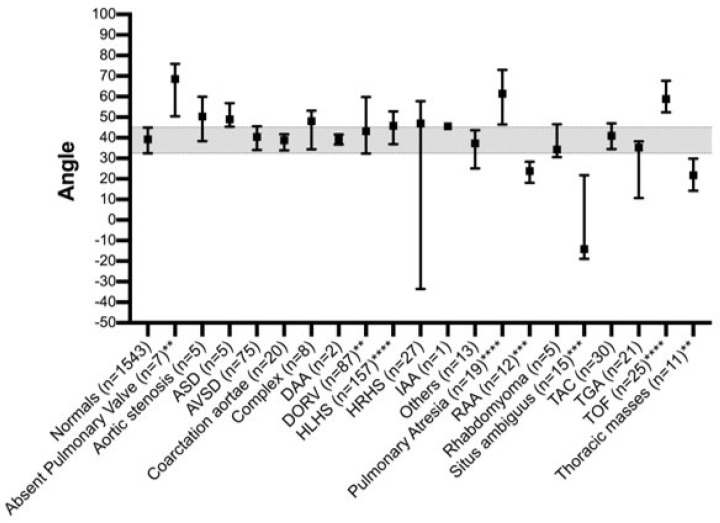
Median and interquartile ranges of the cardiac axis assessed by FINE. The broad grey line indicates the normal fetal cardiac axis is 40–45°. Note that CHDs such as absent pulmonary valve, aortic stenosis, ASD, pulmonary atresia, TOF, and complex CHD display a cardiac axis of >45°. In CHD such as RAA, situs ambiguus the heart axis appears to be <20°. CHDs such as HRHS, situs ambiguous, and TGA display a high range of the cardiac axis. Kruskal–Wallistest with multiple comparisons vs. normal, significance with adjusted *p*-values: HLHS, pulmonary atresia, TOF *p*-value < 0.0001; RAA, situs ambiguus *p*-value = 0.0001–0.001; absent pulmonary valve syndrome, DORV, thoracic masses 0.001–0.01.

**Figure 2 jcm-12-06371-f002:**
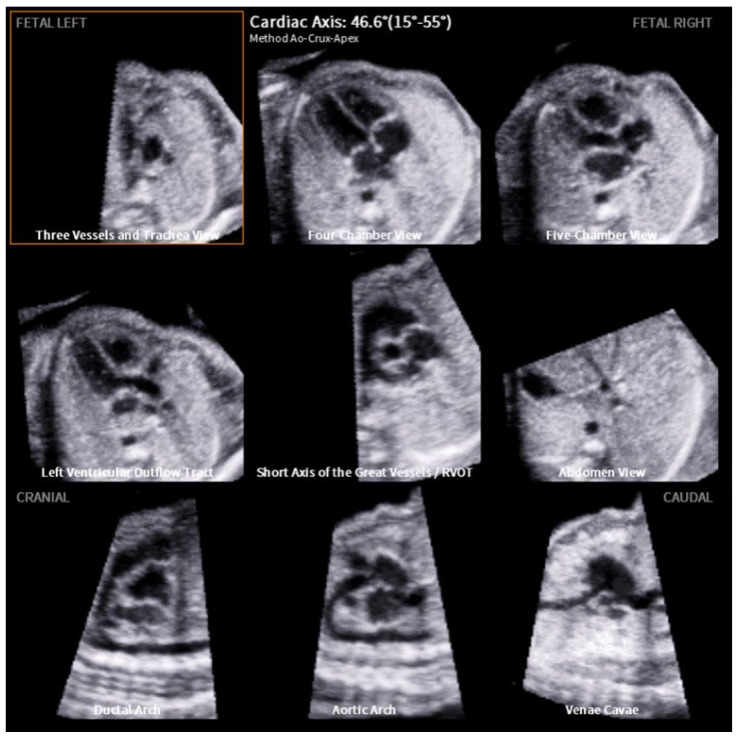
Example of fetal intelligent navigation echocardiography (FINE): a normal fetal heart with a computerized cardiac axis of 46.6°. All nine planes are depicted (from left to right): three vessels and trachea view, four-chamber view, five-chamber view, left ventricular outflow tract, the short axis of the great vessels (right ventricular outflow tract), abdomen view, ductal arch, aortic arch and venae cavae.

**Figure 3 jcm-12-06371-f003:**
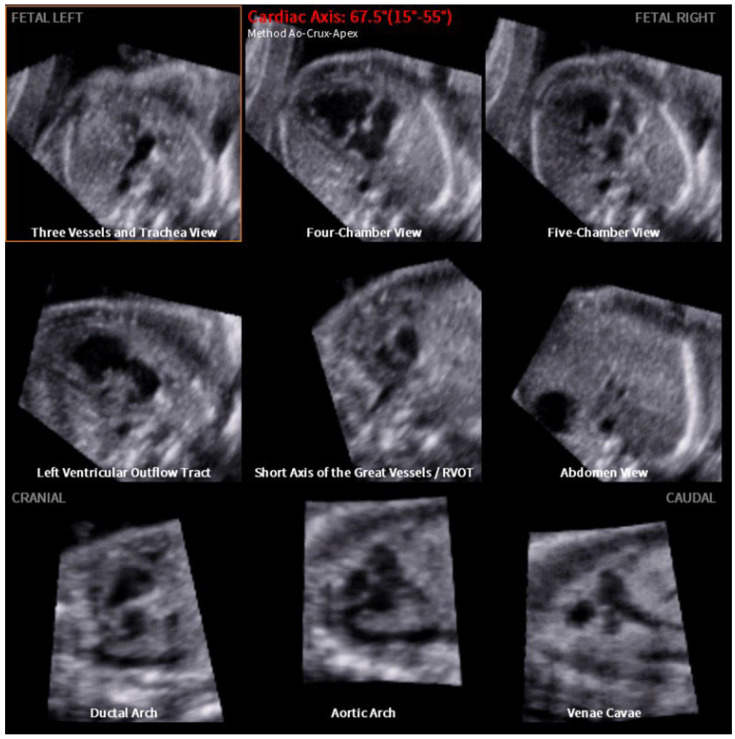
Example of fetal intelligent navigation echocardiography (FINE): the images display a fetus’s heart with a double inlet left ventricle (DILV) and pulmonary valve atresia, with a computerized cardiac axis of 67.5°. All nine planes are depicted (from left to right): three vessels and trachea view—with an absent truncus pulmonalis, four-chamber view—with the absence of the ventricular septum, five-chamber view, left ventricular outflow tract, the short axis of the great vessels (right ventricular outflow tract)—with an absent truncus pulmonalis, abdomen view, ductal arch—which does not become visible, aortic arch and venae cavae.

**Figure 4 jcm-12-06371-f004:**
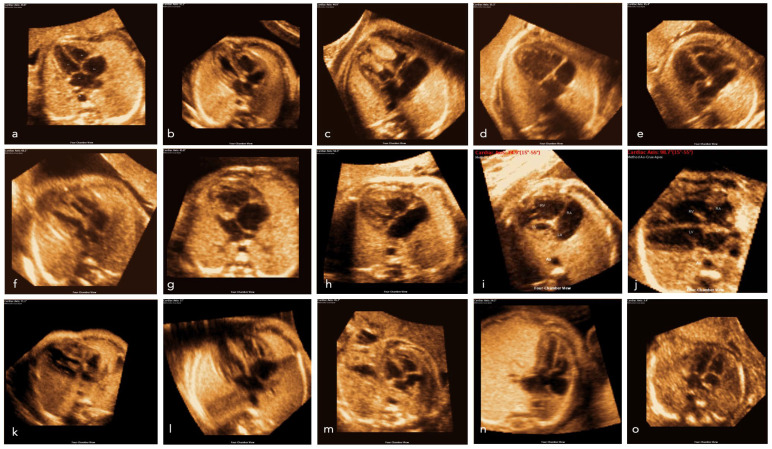
Images of the four-chamber view of a normal fetal heart (**a**) and examples of CHD and thoracic abnormalities from our cohort (**b**–**o**). (**b**) Coarctation aortae, (**c**) rhabdomyoma, (**d**) DILV, (**e**) Ebstein, (**f**) coarctation aortae, (**g**) HLHS, (h) AVSD, (**i**) HLHS, (**j**) tetralogy of Fallot, (**k**) HRHS, (**l**) RAA, m: diaphragmatic hernia, (**n**) CPAM, (**o**) situs ambiguus (heterotaxy).

**Table 1 jcm-12-06371-t001:** Distribution of the fetal cardiac axis measurements in fetuses with CHD/thoracic masses using FINE. Kruskal–Wallis test showed a significant deviation from the normal fetal heart axis in cases of HLHS, pulmonary atresia, TOF (**** *p*-value < 0.0001), RAA, situs ambiguus (*** *p*-value = 0.0001–0.001) absent pulmonary valve syndrome, DORV, thoracic masses (** *p*-value = 0.001–0.01).

CHD/Thoracic Mass	n	Axis Mean (Range)	Axis Median
Absent pulmonary valve **	7	64.0° (38.5–76.2°)	68.6°
Aortic stenosis	5	49.4° (34.2–60.1°)	50.3°
ASD	5	50.6° (45.3–57.4°)	48.9°
AVSD	75	40.0° (19.5–66.9°)	40.5°
Coarctation aortae	20	39.2° (30.7–55.0°)	38.8°
Complex CHD	8	47.1° (22.9–78.2°)	48.1°
DAA	2	39.2° (36.8–41.7°)	39.2°
DORV **	87	45.9° (8.7–105.4°)	43.2°
HLHS ****	157	45.1° (10.2–87.2°)	45.8°
HRHS	27	32.1° (−54.8–65.7°)	49.6°
IAA	1	45.6°	45.6°
Others	13	34.8° (19.9–46.2°)	37.3°
Pulmonary Atresia ****	19	60.2° (26.6–97.0°)	61.5°
RAA ***	12	24.1° (15.3–39.0°)	23.8°
Rhabdomyoma	5	37.7° (28.3–58.5°)	34.2°
Situs ambiguus	15	−8.0° (−31.4–36.7)	−16.8
TAC ***	30	50.3° (24.4–65.7°)	51.6°
TGA	21	34.6° (−3.7–69.6°)	34.8°
TOF ****	25	56.6° (29.4–86.9°)	58.9°
Thoracic masses	11	23.3° (13.4–48.9°)	21.8°
Total	545		

## Data Availability

Our research data are available on request.

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
