# Peer review of "Semi-Automatic Measurement of Fetal Cardiac Axis in Fetuses with Congenital Heart Disease (CHD) with Fetal Intelligent Navigation Echocardiography (FINE)"

_jcm, 2023, doi:10.3390/jcm12196371_

Round 1

Reviewer 1 Report

Dear authors of the work jcm-2645868, I make some constructive observations on your work.

In the results section, they include that they found 15 cases of situs ambiguus, however, visceral situs refers to the position of the viscera in the body, although it implies some type of heart disease, it should not be included as heart disease, I suggest including it. at the end of heart disease.

In the discussion section, the abbreviation “FINE” was already described previously

The FINE technique in the "instroduction" section had been defined as semi-automated, however in the "discussion" section FINE is defined as automated. Please define the procedure correctly.

Conclusion section.

It seems to me that text 251-255 is a discussion of work. Please conclude your observations forcefully.

Author Response

Dear reviewer,

Thank you so much for your time and work you put into reviewing the MS. Here are our answers:

In the results section, they include that they found 15 cases of situs ambiguus, however, visceral situs refers to the position of the viscera in the body, although it implies some type of heart disease, it should not be included as heart disease, I suggest including it. at the end of heart disease. - our situs ambiguus cases were Heterotaxy fetuses -> we added more information on this and stated that situs ambiguus is not simply a CHD

In the discussion section, the abbreviation “FINE” was already described previously -> thanks for this, we changed that part accordingly

The FINE technique in the "introduction" section had been defined as semi-automated, however in the "discussion" section FINE is defined as automated. Please define the procedure correctly. - > it's semi-automated as there are still a few things and adjustment the sonographer need to do - we changed automated to semi-automated

Conclusion section.

It seems to me that text 251-255 is a discussion of work. Please conclude your observations forcefully.

-> thank you for this valuable remark. Obviously there was some information missing. We were able to write a stronger conclusion

Reviewer 2 Report

The authors of the manuscript present results from an original clinical study, by which they investigate the potential diagnostic role of a new echocardiographic modality - Fetal Intelligent Navigation Echocardiography (FINE). They have applied this technique to 545 fetuses affected by CHD/ thoracic masses and 1543 healthy fetuses analyzing the cardiac axis of these fetuses and relating the findings to the presence/abcence of congenital structural heart changes. According to the presented results in fetuses with CHD the cardiac axis deviates significantly from the normal range, so FINE appears to be a promising and reliable tool for early diagnosis of congenital heart defects.

The topic of the article is curent and the results, presented by the authors - important from a clinical and scientific point of view. The very manuscript is generally well-structured and readable. 

I have the following recommendations to the authors:

1. I suggest that the authors modify the title of their article, for instance "Semi-automatic measurement of cardiac axis with Fetal Intelligent Navigation Echocardiography (FINE) in fetuses with congenital heart disease" or something like that.

2. Lines 32-33: The 2nd sentence is missing a reference.

3. The Introduction should be more succinct - I suggest that the authors cut much of the general, well-known information about echocardiography and focus/leave the information that is new and directly related to the article topic/title. Some parts of the text could be moved to Discussion.

4. Methods for statiscial analysis of the collected data are not described in "Materials and Methods"

5. Please, add "Study limitations" at the end of "Discussion" 

Author Response

Dear Reviewer,

Thank you so much for putting in the time and the effort you put in reviewing our MS

  1. I suggest that the authors modify the title of their article, for instance "Semi-automatic measurement of cardiac axis with Fetal Intelligent Navigation Echocardiography (FINE) in fetuses with congenital heart disease" or something like that.-> this is a great suggestion, we were able to figure out a much more suitable headline

2. Lines 32-33: The 2nd sentence is missing a reference. -> this was added

3. The Introduction should be more succinct - I suggest that the authors cut much of the general, well-known information about echocardiography and focus/leave the information that is new and directly related to the article topic/title. Some parts of the text could be moved to Discussion. -> we were able to adjust the introduction 

4. Methods for statiscial analysis of the collected data are not described in "Materials and Methods" -> those were added in detail

5. Please, add "Study limitations" at the end of "Discussion" - this important aspect was added to the MS